# What Is the Entropy of a Social Organization?

**DOI:** 10.3390/e21090901

**Published:** 2019-09-17

**Authors:** Christian Zingg, Giona Casiraghi, Giacomo Vaccario, Frank Schweitzer

**Affiliations:** Chair of Systems Design, ETH Zurich, Weinbergstrasse 58, 8092 Zurich, Switzerland

**Keywords:** multi-edge network, network ensemble, Shannon entropy, social organization

## Abstract

We quantify a social organization’s potentiality, that is, its ability to attain different configurations. The organization is represented as a network in which nodes correspond to individuals and (multi-)edges to their multiple interactions. Attainable configurations are treated as realizations from a network ensemble. To have the ability to encode interaction preferences, we choose the generalized hypergeometric ensemble of random graphs, which is described by a closed-form probability distribution. From this distribution we calculate Shannon entropy as a measure of potentiality. This allows us to compare different organizations as well as different stages in the development of a given organization. The feasibility of the approach is demonstrated using data from three empirical and two synthetic systems.

## 1. Introduction

Social organizations are ubiquitous in everyday life, ranging from project teams, e.g., to produce open source software [1], to special interest groups, such as sports clubs [2] or conference audiences [3] discussed later in this paper. Our experience tells us that social organizations are highly dynamic. Individuals continuously enter and exit, and their interactions change over time. Characteristics like these make social organizations complex and difficult to quantify.

Network science allows studying such complex systems in terms of networks, where nodes represent individuals and edges their interactions [4,5]. Under this assumption, a social organization can be represented by a network ensemble. Every network in this ensemble corresponds to one possible configuration of interactions in this organization. Thus, the network that we can reconstruct from observed interactions is only one particular realization from this ensemble. Other configurations can also be realized with a given probability. To cope with this, we need a probability distribution that characterizes the network ensemble and reflects its constraints, such as given numbers of nodes or interactions, or preferences for interactions. The probability space defined by this distribution can then be seen as a way to quantify the “number” and the “diversity” of possible states of a social organization. We argue that such possible states give an indication of the “potentiality” of this organization, i.e., its ability to attain different configurations.

But how can the potentiality be measured? First, we need to decide about a probability distribution suitable for reflecting the interactions and constraints in social organizations. Second, based on this distribution, we need to quantify the diversity of the organization. To solve the first task, in this paper we utilize the hypergeometric ensemble, as explained in Section 3.1. To solve the second task, we compute the information entropy of this ensemble, as shown in Section 3.2.

Information entropy [6] has recently gained popularity in the study of social organizations. Shannon entropy has, for example, been applied to study communication networks [7], human contact events [8], or anomalies in computer networks on a campus [9]. By generalizing the concept of entropy, or even complexity measures [10], classifications for complex systems [11] have been suggested. Finally, entropies have also been applied in combination with network ensembles to analyze complexity behind structural constraints [12] or spatial constraints [13] or how restrictive topological constraints are [14].

Recent works that combine network ensembles and entropy analyze the effect of predefined constraints. For example, in the work by the authors of [14], the author studies how entropy changes when fixing degree sequences or community structures, that are derived from the network topology. By enforcing such topological constraints, the resulting ensembles serve as null models to study the expected level of order given the fixed constraints. However, real systems are affected by a very large number of constraints and because they are so many, a list stating all of them one by one is unfeasible. Instead, we focus on their combined effect, which we extract by applying the generalized hypergeometric ensemble, gHypEG [15], to a given network representation. This allows the encoding of observed interaction preferences among every pair of individuals as biases in the edge formation. By this method, we capture in the network ensemble the combined restriction of all constraints that manifest as interaction preferences between individuals.

Specific structural constraints can be measured for example by network measures such as modularity or nestedness. What we propose is a measure for the combined effect of such constraints in order to capture the potentiality of the analyzed organization. Clearly with our measure we can not consider the origin of individual constraints. However, our measure provides a description of the whole organization and how constrained it is overall.

The paper is organized as follows. In Section 2 we derive our measure of potentiality for a social organization based on the Shannon entropy of a probability distribution. In Section 3 we first explain how this probability distribution can be calculated for a Generalized Hypergeometric Ensemble. Then, we also show how to obtain the Shannon entropy of this distribution by means of a limit-case approximation, because direct computations are infeasible because of the large size of the ensemble. In Section 4, we measure the potentiality of 3 empirical social organizations and then compare the computed values across the organizations. Finally, in Section 5, we summarize our approach and comment on its strengths and weaknesses.

## 2. Quantifying the Potentiality of a Social Organization

### 2.1. Network Representation of a Social Organization

We adopt a network perspective to study social organizations. The nodes of the network represent individuals, and the observed interactions between them are represented by edges. If multiple interactions between the same pair of individuals occur, we consider them as separate edges, or so-called multi-edges [16]. For simplicity, we will always refer to them as edges. In this article, we will focus on undirected edges without self-loops; however, the methodology discussed can easily be extended to encompass directed edges and self-loops.

According to this perspective, the observation of a social organization composed of *n* individuals yields an network $\hat{g}$, with *n* nodes and *m* edges, where *m* is the number of observed interactions. The “state” of the social organization, instead, is defined by a network ensemble composed of all possible networks $S=\{g_{0},\;\ldots ,\;g_{N}\}$, which encompass all possible configurations the social organization could attain with $\hat{g}\in S$.

As an example, Figure 1 illustrates every possible network for three nodes and an increasing number of edges *m*. Whereas for three nodes and two edges there are six possible networks, for three edges already 10 networks result. For 10 nodes and 10 edges there would be more than $2\times 10^{10}$ possible networks. The general expression for the number of possible networks is
(1)n(n−1)2+m−1m
where n(n−1)/2 denotes the number of combinations between *n* nodes. Equation (Equation 1) can be derived directly from the known formula for drawing unordered samples with replacement. The replacement is important because we consider multi-edges.

Notwithstanding the large number of possible networks, not all of them appear with the same probability. *g* represents a particular network and P(g) represents the probability to find *g*, given *n* and *m*. A proper expression for P(g) has to take into account that the ensemble, in addition to a fixed number of nodes and edges, also may have other constraints that need to be reflected in the probability distribution. This issue will be further discussed in Section 3. However, assuming that we have such an expression for P(g), the information from this can be compressed by calculating Shannon entropy [17]:(2)$$H=-\sum_{g\in S}P\left(g\right)\log P(g)$$
where *S* denotes the set of all possible networks for fixed *n* and *m*.

### 2.2. Potentiality of a Social Organization

#### 2.2.1. Potentiality and Constraints

In our network representation, a large number of possible networks translates into a large number of possible configurations that can be attained by the social organization. Thus, we can use entropy to characterize the potentiality of the social organization, that is, its ability to attain these different configurations under the existing constraints. These constraints limit the number of configurations, i.e., they reflect that a social organization cannot change from a given configuration to any arbitrary other configuration. Thus, constraints lower the potentiality of social organizations.

Such constraints can be temporal, i.e., they impose an order of occurrence to the edges in the network, as extensively examined in the works by the authors of [8,18]. Or there can be spatial constraints that restrict the individuals in the choice of communication partners [19,20]. Social organizations can also be subject to hierarchical constraints [21], restricting, e.g., the flow of information, or to social constraints [22], as discussed in Section 4.

#### 2.2.2. How to Proxy Constraints

We consider distributions P(g) that capture communication biases among the individuals. These biases, or preferences, are the consequences of the constraints that restrict the social organization. We take the observed number of interactions between each pair of individuals in a defined time interval as the proxy for the constraints. For this reason, we set the expected number of interactions between each pair of nodes in the ensemble to the observed ones. This choice ensures that the distribution P(g) encodes the constraints in the ensemble, because we assume that constraints are expressed in the number of interactions between the nodes.

In the next Section, we will demonstrate how to specify the probability distribution P(g) characterizing the network ensemble such that this is achieved. To do so, we will employ the generalized hypergeometric ensemble (gHypEG) developed by Casiraghi and Nanumyan [15].

#### 2.2.3. Network Ensembles and Their Probability Distribution

What have we obtained by calculating Shannon entropy, i.e., a single number to characterize P(g)? To fully understand this, we have to recapture what information the probability distribution actually contains. P(g) in fact characterizes the “diversity” of potential networks, i.e., the possible network configurations that can appear under the given constraints encoded in P(g). We denote the totality of these configurations as the network ensemble. If there are only a few network configurations possible, the ensemble is comparably small and the resulting entropy is low. On the other hand, if many network configurations are possible, the ensemble becomes very large and the entropy is high.

## 3. Introducing the Generalized Hypergeometric Ensembles

### 3.1. Obtaining P(g)

For the calculation of Shannon entropy, Equation (Equation 2), we implicitly assume that P(g) is known. There are mainly two candidates for P(g) that fit our requirements. One is the family of exponential random graphs, also known as ERGMs [23,24]. ERGMs follow an exponential distribution, thus it is possible to compute their Shannon entropy. Moreover, they can incorporate a broad set of properties and constraints [25], which can fit virtually any characteristics of observed networks. However, ERGM fitting algorithms, especially when fitted to multi-edge networks, tend to not converge and thus cannot be efficiently computed for large networks.

Additionally, they are intended to consider a predefined set of constraints. However, predefining all constraints of a social organization is unfeasible, given the very large number of constraints. Existing applications of ERGMs therefore examine specific constraints, as for example by fixing a clustering coefficient or degree assortativity [26]. However, we intend to measure the combined effect of the constraints, and therefore our choice is the second candidate, which is the generalized hypergeometric ensemble of random graphs [15,27] (gHypEG). This ensemble extends the configuration model (CM) [28] by encoding complex topological patterns, while at the same time preserving expected degree sequences.

Specifically, gHypEG keeps the number of nodes and edges fixed. However, different from the CM, the probability to connect two nodes depends not only on their (out- and in-) degrees (i.e., number of stubs), but also on an independent propensity of the two nodes to be connected, which captures non-degree-related effects as explained in the following.

#### 3.1.1. Parameters of a gHypEG

The distribution of networks in a gHypEG is formulated in terms of two sets of parameters. The first set of parameters is represented in terms of the combinatorial matrix Ξ that encodes the CM. This means the entries $\Xi_{ij}$ reflect all ways in which nodes *i* and *j* can be linked. As will be explained later in an undirected network without self-loops, this number is $2{\tilde{d}}_{i}{\tilde{d}}_{j}$ for rescaled degrees ${\tilde{d}}_{i}$ and ${\tilde{d}}_{j}$ of nodes *i* and *j*.

The second set of parameters is represented in terms of the propensity matrix Ω, which encodes preferences of nodes to be connected. That means, propensities allow to constrain the configuration model such that given edges are more likely than others, independently of the degrees of the respective nodes. This creates a bias which is expressed by the ratio between any two elements $\Omega_{ij}$ and $\Omega_{kl}$, i.e., the odds ratio of observing an edge between nodes *i* and *j* instead of between *k* and *l*.

The matrices Ξ and Ω both have dimension n×n, where *n* is the number of nodes. The probability distribution that reflects the biased edge allocation described above is given by the multivariate Wallenius noncentral hypergeometric distribution [29]. I.e., the probability of a network *g* in the gHypEG with parameters Ξ and Ω is given as follows,
(3)P(g|Ξ,Ω)=∏i,j∈V,i<jΞijAij∫01∏i,j∈V,i<j1−zΩijSΩAijdz
with
(4)$$S_{\Omega }=\sum_{i,j\in V,i<j}\Omega_{ij}\left(\Xi_{ij}-A_{ij}\right).$$

Equations (Equation 3) and (Equation 4) hold for undirected networks without self-loops (i<j).

#### 3.1.2. Calculating Ξ for Networks

We obtain the Ξ matrix for a given network according to Definition 4 and Lemma 3 in the work by the authors of [15]. But, since in our applications there are no self-loops, we implement additional correction factors to preserve the expected degrees in the ensemble. Specifically, we ensure that the expected degrees are equal to the degrees in the initial network. The details can be found in Appendix A. Our $\Xi_{ij}$ are therefore
(5)$$\Xi_{ij}:=\left\{\begin{array}{cc} 2\left(d_{i}\theta_{i}\right)\left(d_{j}\theta_{j}\right) & \mathrm{if}\;i<j\\ 0 & \mathrm{else} \end{array}\right.$$
where $d_{i}$ and $d_{j}$ denote the degree of nodes *i* and *j*, and $\theta_{i}$ and $\theta_{j}$ denote the correction factors that ensure the expected degrees are preserved. In this definition, the diagonal elements are 0 because we do not allow for self-loops. Also the entries in the lower triangular part are 0 to account for the networks being undirected.

#### 3.1.3. Calculating Ω for Networks

We obtain the respective Ω matrix for a given Ξ matrix according to Corollary 7.3 in [15]. Thereby we ensure that, in addition to the expected degrees, even the expected numbers of edges between all pairs of nodes in the ensemble are equal to the respective numbers of edges in the initial network. Hence, our $\Omega {}_{ij}$ are
(6)$$\Omega_{ij}:=\left\{\begin{array}{cc} \frac{1}{c}\log\left(1-\frac{A_{ij}}{\Xi_{ij}}\right) & \mathrm{if}\;i<j\\ 0 & \mathrm{else} \end{array}\right.$$
where $A_{ij}$ is the number of edges between nodes *i* and *j*, and *c* is a multiplicative constant which we choose such that the values in Ω are between 0 and 1 for simplicity. We refer to [15] for how special cases such as $A_{ij}=\Xi_{ij}$ can be handled. Again, the entries on the diagonal and in the lower triangular part of Ω are 0 to account for the networks having no self-loops and being undirected.

### 3.2. Tractability of the Entropy

#### 3.2.1. Multinomial Entropy Approximation

Computing the Shannon entropy of the fitted gHypEG according to Equations (Equation 2)–(Equation 4) is not straightforward because of the very large number of networks in this ensemble. If we were to simply plug the probabilities of all networks into Equation (Equation 2), the very large number of summands quickly becomes infeasible. Thus, instead of literally computing the entropy for a fitted gHypEG, we compute *H* using the fact that, for large networks, the multinomial distribution approximates the Wallenius distribution. The details of the derivation can be found in Appendix B. Hence, the gHypEG entropy can be approximated as
(7)Hmult=−log(m!)−m∑i,j∈V,i<jpijlog(pij)+∑x=2m∑i,j∈V,i<jmxpijx(1−pij)m−xlog(x!)
where *m* is the number of edges in the network, *V* is the set of nodes, and
(8)$$p_{ij}=\frac{\Xi_{ij}\Omega_{ij}}{\sum_{kl}\Xi_{kl}\Omega_{kl}}$$

#### 3.2.2. Computing the Multinomial Entropy

Equation (Equation 7) can be computed efficiently even for large ensembles. In SciPy [30], there exists an efficient implementation for computing the entropy of a given multinomial distribution. Our contribution is to apply this to approximate the entropy for a given gHypEG defined by Equations (Equation 7) and (Equation 8).

### 3.3. Comparing Entropy Values

#### Normalizing Value Ranges

The value range of Equation (Equation 7) depends on the number of nodes *n* and edges *m*. In particular, it is a known fact that Shannon entropy attains its maximum value $H^{\max}$ at equiprobability [17]. Thus, the entropy values are always in the interval $\left[0,H^{\max}\right]$.

For undirected networks without self-loops equiprobability corresponds to
(9)$$p_{ij}^{\max}=\frac{2}{n(n-1)}$$
i.e., all possible pairs of nodes can be chosen with the same probability. For two different ensembles, however, $H^{\max}$ can be different because it depends on *n* and on *m* via Equation (Equation 7). To compare the values of $H^{\mathrm{mult}}$ (Equation (Equation 7)), we normalize them by their respective maximum values:(10)$$H^{\mathrm{norm}}:=\frac{H^{\mathrm{mult}}}{H^{\max}}\equiv \hat{H}\in \left[0,1\right]$$

A small value means that the ensemble contains only very few networks, given the constraints. With respect to the $p_{ij}$, this means that only very few have probabilities considerably different from zero. A large value, on the other hand, means that pairs of nodes are chosen almost at random, because of the very few constraints. Hence, $\hat{H}$ indeed reflects the potentiality of the social organization, namely its ability to attain different configurations under given constraints.

### 3.4. Examples for $\hat{H}$

#### 3.4.1. Two Special Cases

To illustrate how constraints can be encoded in the ensemble, we use two examples, a complete network and a star network (see Figure 2) for which we consider undirected edges and no self-loops. We fit the Ξ and Ω matrices according to Equations (Equation 5) and (Equation 6).

#### 3.4.2. Complete Network

This network has 10 nodes and 90 edges. Because we consider a multi-edge network, each node has two edges to every other node. This results in $d_{i}=d_{j}=18$. The correction factors are obtained by solving the set of equations in the Appendix A and yield $\theta_{i}=\theta_{j}=1.054$. The resulting network is depicted in Figure 2 (left), and the resulting Ξ and Ω matrices are stated in full in Appendix C. The entries of the Ξ matrix for this network are computed according to Equation (Equation 5) as
(11)$$\Xi_{ij}^{C}=\left\{\begin{array}{cc} 720 & \mathrm{if}\;i<j\\ 0 & \mathrm{else} \end{array}\right.$$

Although in the complete network every node has the same number of edges to every other node, there are no preferences for specific pairs of nodes. Therefore, one way to choose the Ω matrix to encode no bias is
(12)$$\Omega_{ij}^{C}=\left\{\begin{array}{cc} 1 & \mathrm{if}\;i<j\\ 0 & \mathrm{else} \end{array}\right.$$
which corresponds to Theorem 8 in the work by the authors of [15] for an undirected network without self-loops. Remember that $\Omega_{ij}/\Omega_{kl}$ is the odds ratio of observing an edge between nodes *i* and *j*, instead of nodes *k* and *l*. By choosing $\Omega_{ij}$ according to Equation (Equation 12) such ratios are always equal to 1. By plugging $\Xi^{C}$ and $\Omega^{C}$ into Equation (Equation 10) we obtain $\hat{H}=1$. This means that there are no edge preferences between particular pairs of nodes, which is trivial because the example was chosen as such.

#### 3.4.3. Star Network

This network has again 10 nodes and 90 edges. But this time there is one center node and nine peripheral nodes, i.e., the network has the constraint that each peripheral node has 10 edges that are all attached to the center node, as depicted in Figure 2 (right). This results in a degree $d_{i}=90$ for the center node placed at i=1 and degrees $d_{j}=10$ for all peripheral nodes j≠1. Again, the Ξ and Ω matrices for this network are stated in full in Appendix C.

When computing the Ξ matrix according to Equation (Equation 5), we obtain
(13)$$\Xi_{ij}^{S}=\left\{\begin{array}{cc} 3592 & \mathrm{if}\;i=1,j>1\\ 2 & \mathrm{if}\;i>1,j>i\\ 0 & \mathrm{else} \end{array}\right.$$
where the center node is placed at i=1.

The other matrix, Ω, has to reproduce the constraint that peripheral nodes can only communicate with the center node. One choice to fulfil this is
(14)$$\Omega_{ij}^{S}=\left\{\begin{array}{cc} 1 & \mathrm{if}\;i=1\;\mathrm{and}\;i<j\\ 0 & \mathrm{else} \end{array}\right.$$

This choice of Ω specifies that observing an edge from node 1 (the center) to any two peripheral nodes *k* or *l* occurs with the same probability, because the odds ratio $\Omega_{1k}/\Omega_{1l}$ is equal to 1. On the other hand, the odds ratio for a peripheral node *i* to form a link with another peripheral node *k* instead of with the center node 1, namely, $\Omega_{ik}/\Omega_{1i}$, is 0 (or infinity if the inverse ratio is formed). This encodes the constraint that all edges have to be incident to the center node.

By plugging $\Xi^{S}$ and $\Omega^{S}$ into Equation (Equation 10) we obtain $\hat{H}=0.27$ for our star network. This value is much lower than for the complete network and reflects the very restrictive constraint that all edges have to be incident to the center node.

## 4. Applications to Real-World Datasets

### 4.1. Examined Datasets

In this Section we apply our potentiality measure to five empirical networks of social organizations. These networks were constructed from publicly available datasets which we shortly describe in the following.

#### 4.1.1. Southern Women Dataset

The Southern Women Dataset was introduced by Davis et al. [31] and contains information about 18 women and their participation in 14 social events. Instead of constructing a bipartite network, we use a so-called one-mode representation (i.e., a specific projection of the bipartite network), in which the women correspond to the nodes and the edges correspond to co-participations in the social events. There are no self-loops in this network and edges are undirected.

#### 4.1.2. Karate Club Dataset

The Karate Club Dataset was introduced by Zachary [2], and the network contains 34 nodes corresponding to the members of this university Karate club. Edges correspond to co-participation of members in different activities. They are all undirected and there are no self-loops. There are 8 activities considered, thus the number of possible edges between any pair of nodes is less or equal than 8. In total, there are 231 edges.

#### 4.1.3. Conference Dataset

The Conference Dataset Dataset is part of the SocioPatterns project. It contains data about interactions among conference participants during the ACM Hypertext 2009 conference. [3] To measure the interactions, participants wore proximity sensors. For each interaction between two participants the measured information contains their anonymous ids as well as the time of the respective measurement. From this information, we constructed three networks, one for each day of the conference. In each network the nodes correspond to the 113 participants in the data and the edges correspond to their interactions at the respective day. None of the networks contain self-loops and all edges are undirected. All three networks have the same set of nodes, but differ slightly in the number of edges as seen in Table 1.

#### 4.1.4. Network Overview

To summarize the networks, Table 1 lists the general network statistics besides the computed potentiality values of $\hat{H}$. Furthermore, all networks are visualized in Figure 3. This Figure already suggests that the networks are structurally different. For example, the Karate Club network shows a cluster structure which is not apparent in the Southern Women network, and all Conference networks have isolated nodes, which neither the Karate Club network nor the Southern Women network have.

### 4.2. Potentiality of the Empirical Networks

For each of the five empirical networks, we computed the potentiality $\hat{H}$ as outlined in Section 2 and Section 3. The computed values for $\hat{H}$ are listed in Table 1. In the following we comment on the results.

#### 4.2.1. Southern Women Network

This network attains a very high potentiality at approximately $\hat{H}=0.9$, meaning that there are only few constraints in the women’s interaction. In fact, there are almost no preferences for specific pairs of women. Instead, everyone interacts with everyone else in a rather homogeneous way. The absence of a preference structure in co-attending events is also visible in the network plot in Figure 3, which looks similar to the complete network considered in Section 3.4. This corresponds to the high density, *D*, in Table 1. Note also that the measured potentiality is high, but it is still not at its maximum value 1, i.e., there are constraints present in the network that restrict interactions. Evidence for these constraints are the two groups that were identified among the southern women in the works by the authors of [31,32].

#### 4.2.2. Karate Club Network

This network results in a lower potentiality at around $\hat{H}=0.3$, which indicates that the network is more restricted by constraints. Indeed, it is known that two social groups, which both had their own leader, coexisted in the Karate club. Most interactions among the club members occurred within the groups and, in particular, with the respective group leaders. These restrictions explain why the potentiality of this network is not particularly large, especially when compared to the Southern Women network.

#### 4.2.3. Conference Networks

The lowest potentialities are attained at ~0.2 by the networks of the conference participants. For each network we observed that the nodes had high degrees, because of the multi-edges, but were linked only to a few other nodes (i.e., a rather sparse network). This implies a relatively strong preferential linkage between specific pairs of nodes. On the other hand, on all three days of the conference, there was at least one individual who communicated with at least 50% of conference participants in the dataset (probably the conference organizer). This induces a star-like interaction effect. However, there were also a smaller number of isolated nodes which were, on a given day, not involved in any interaction.

Isolated nodes decrease the potentiality, because in the theoretical maximum entropy $H^{\max}$ they have to be considered. When omitting these isolated nodes, we still find only slightly higher values of the potentialities around 0.24 because of their small number. Furthermore, it is remarkable that the networks of all three days have similar potentialities. One could have expected that on day 3 of the conference, participants mainly interact with those they already know. However, this is obviously not the case.

## 5. Conclusions

In this paper, we address the question how many different states a social organization can attain. Why is this of importance? We argue that the number of such possible states is an indication of the ability of the organization to respond to various influences. As there can be a vast variety of such influences, the corresponding number ideally should be very large. This indicates that, even for unforeseeable events, the social organization still has many ways to respond. We call such an ability the potentiality of the organization.

To quantify this potentiality, we need an appropriate representation of the social organization. In this paper, we choose a network approach, where nodes represent individuals and edges their repeated interactions. This leads to a multi-edge network. A network ensemble then contains all possible networks that fulfill a given set of constraints. Such constraints are detected from the observed network and encoded as propensities, i.e., as interaction preferences. The statistical ensemble of all possible networks is then given by the generalized hypergeometric ensemble (gHypEG). From this, we can calculate a Shannon entropy, which is used to proxy the potentiality of the organization.

In the following, we comment further on the strengths and weaknesses of our approach.

### 5.1. Fixed Numbers of Nodes and Edges

We focus on ensembles with a fixed number of nodes and edges, hence imposing that only networks of this size are attainable by the organization. Thereby, we neglect system growth on purpose, to provide a general measure of potentiality.

### 5.2. Large Number of Degrees of Freedom

Using gHypEG, we are able to consider the maximum possible degrees of freedom, meaning that every detail is modeled in the Ξ and Ω matrices. This way, we obtain a high model complexity. A more refined approach could be to compare ensembles of various complexities based on goodness-of-fit measures such as AIC or BIC. Thereby also simpler ensembles could be involved that, for example, consider communication preferences only between certain communities in the network. Such choices of simpler ensembles were not considered because, again, we want to provide a general approach not restricted to systems with particular community structures.

### 5.3. Computability

For social organizations with only 10 individuals and 10 interactions there are already more than $2\times 10^{10}$ possible network representations in the ensemble. According to Equation (Equation 2), all of these networks must be considered individually to compute the Shannon entropy. Hence, even simple approaches to directly compute the entropy are computationally infeasible already for very small organizations. Our approach instead uses that the Wallenius distribution underlying the gHypEG converges to a multinomial distribution in the limit of large networks for which the entropy can be computed efficiently. This allows to study the potentiality of a wide range of social organizations.

Our main methodological contribution is indeed the novel way to conceptualize potentiality for a social organization using its representation as a multi-edge network. As long as this representation is justified, our approach can be extended to other systems.

## Figures and Tables

**Figure 1 entropy-21-00901-f001:**
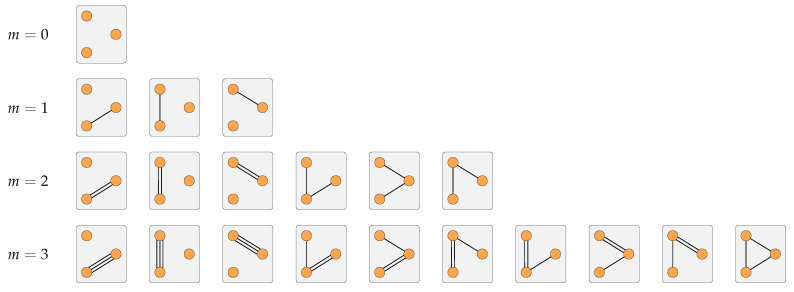
Visualization of the possible networks for 3 nodes and different numbers of edges. The edges are undirected and self-loops are not considered.

**Figure 2 entropy-21-00901-f002:**
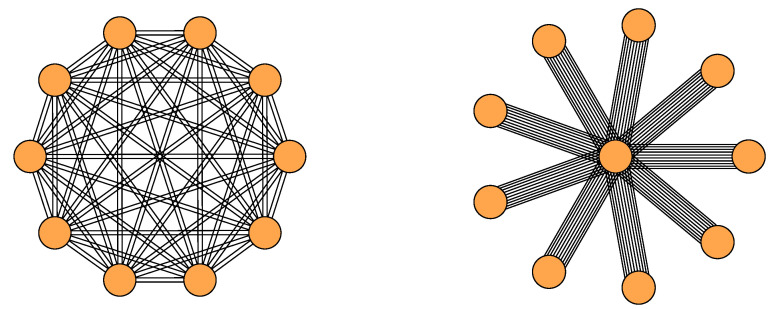
The two networks from Section 3.4. **Left**: Multi-edge complete network. **Right**: Multi-edge star network.

**Figure 3 entropy-21-00901-f003:**
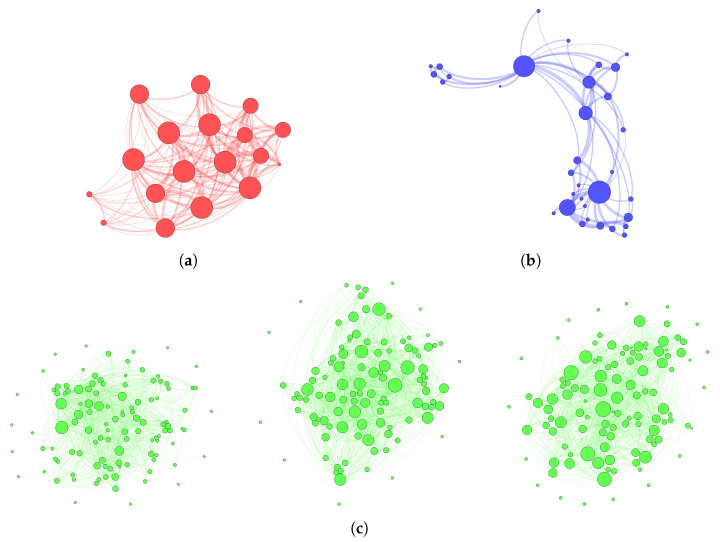
Network visualizations: (**a**) Southern Women (red), (**b**) Karate Club (blue), (**c**) Conference (green) at day 1 (left), day 2 (center), and day 3 (right). The node size is proportional to the degree.

**Table 1 entropy-21-00901-t001:** Network statistics of the 5 examined empirical networks. *n* and *m* denote the number of nodes and edges in each network, respectively. m/n is the average number of multi-edges per node. *D* is the density of the network, i.e., the number of linked node pairs normalized to the total number of possible node pairs, after reducing all multi-edges into single edges. $\hat{H}$ denotes the normalized entropy computed according to Equation (Equation 10). ${\hat{H}}_{\mathrm{gcc}}$ corresponds to $\hat{H}$ when only the largest connected component in each network is considered. All networks are undirected and have no self-loops.

Network	*n*	*m*	m/n	*D*	$\hat{H}$	${\hat{H}}_{\mathrm{gcc}}$
Southern Women	18	322	17.89	0.91	0.89	0.89
Karate Club	34	231	6.79	0.14	0.31	0.31
Conference t=1	113	6925	61.28	0.15	0.21	0.24
Conference t=2	113	7131	63.11	0.17	0.22	0.25
Conference t=3	113	6762	59.84	0.15	0.19	0.23

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
