# Peer review of "What Is the Entropy of a Social Organization?"

_entropy, 2019, doi:10.3390/e21090901_

Round 1

Reviewer 1 Report

I think this paper is interesting and worth to be published. The concept of entropy of a network is not really new but the treatment and results presented are interesting and may stimulate further work. The paper is clearly written and easy to understand. the methodology is sound and the examples presented are interesting. I suggest publication

Reviewer 2 Report

Dear authors, 

I am sorry, but due to several evident flaws, I reject the publication of the manuscript. 

More in details, the research question is badly overstated: indeed, social networks are known to display strong community structures, that is links are not uniformly distributed in the graph. In the most common community detection algorithms, such a structure is detected by the modularity, which compares the observed intra-community edges with those expected by a null-model constraining the degree sequence. Already this quantity carries more information respect to the relative entropy presented by the authors. Do not misinterpret me: I am not stating that the modularity is the best target function ever, neither that the entropy of the system should not capture the modular structure of a social network. I am saying that the actual measure, beside being interesting on its own, is not that new and other approaches have been proposed during the years. I will not say that the need of such a measure is an "open problem", but that there are several possible interpretations of the same issue. In the whole paper the authors stress the novelty of the proposal, which is strongly overstated, in my view. 

Then, there is the problem of the null-model. The authors say (line 44): «However, real systems are affected by a very large number of constraints and because they are so many, they cannot be easily predefined. To consider also these constraints, we use the generalized hypergeometric ensemble, gHypEG [15]». That is not true: at the end of the day, gHypEG consider the strength sequence and assume that (multi-)edges are uniformly distributed, thus it put constraints on the null-model and, luckily, it is clear what they are. Beside this, it is much safer to know which are the information that the null-model is considering than saying «We are considering everything, even something we are not aware of», that is what a reader is guessing from your words. 

At line 125, the authors say that ERGs are not suited for every kind of constraints: actually, it is just partially true. Indeed in [1] the authors provide a proper strategy of how to explore the network ensembles where the constraints are more complicated that the ordinary degree sequence, strength sequence or any other linear quantities in the adjacency matrix. Indeed it is time consuming, probably not so elegant and not needed when the constrained patterns have no obvious interpretation, but still there are some proposals in the literature regarding how to encode non trivial constraints. Beside this, the overall literature review is not sufficient to describe in details the efforts in the definition of entropy-based null-models for the analysis of complex networks. 

Finally, when discussing the applications of their method to real systems, the authors focus on some monopartite social networks and the weighted projection of the bipartite network of Southern Women. I would not consider the latter application, since the original bipartite structure of the network (which has a 2 nested blocks structure) is carrying more information regarding the system than its projection. Indeed, in this case the "normalised entropy" signal is quite low and I suspect that it is due to the projection.

For all those reasons, I think that the manuscript is not suitable for publications. 

Nevertheless, I would like to reassure the authors: the idea is interesting, even if not revolutionary. I thus suggest the authors to rewrite the paper, add other comparisons with other measures already present in the literature and other null-models (not necessarily [1] which is indeed time consuming and probably not needed). Anyway, the most important thing is to avoid uselessly overstating results since it really bothers the reader.

Best regards.

p.s. There is a typo in (A11) and (A12): the factorial is missing.

[1] Fischer et al, PRL 115, 188701 (2015)

Reviewer 3 Report

The manuscript by Zigg et al. proposed to evaluate the social network potentiality using the entropy of hyper-geometric network ensembles. The ensemble are here carefully defined and justified and their entropy is evaluated by performing controlled approximation. This entropy is taken as a measure for social network potentiality and the framework is applied on two benchmark networks and on several real datasets.

The work is well written and I am sure it will be of interested to the network science community and to social scientists.

I therefore judge it suitable for publication in Entropy in its current form.

Round 2

Reviewer 2 Report

Dear authors, 

essentially there were 3 problems that I saw in the previous version:

the paper overstated the novelty of the proposal; there was a general lack in the comparison of the findings with results from other proposals;  there was a general lack in the comparison of the findings using different null models. 

Before going into more details, I have to remark that even if it is not a review paper, in the moment you are proposing a new measure is indeed mandatory to compare your proposal with others in the literature, otherwise there is no room to get the strong and weak points of your, as well as other, measures. 

The authors, in the actual version of the manuscript (weakly) addressed only the first point and that is not enough, thus I again reject the publication of the manuscript.

In the interest of the authors, I comment their answers.

We appreciate that you emphasize the need for the comparison with other measures. In our revised manuscript we rewrote parts of the introduction and added a paragraph in which we discuss the advantages of our measure in relation to others for this application. Also, we would like to emphasize that we intend to capture constraints that go beyond communities. For example, the hierarchical structure of a company constrains (beyond the team division) how individuals talk to specific other individuals.

Actually, such a comparison is present only in the Introduction. Beside this, modularity was just the first idea that came to my mind, a more detailed discussion is welcome. Moreover, it is not clear how your measure goes beyond the community structure.

In our approach we describe the system as a whole, which is complementary to existing approaches that model how a particular aspect restricts the flexibility of the system. We have to avoid manually specifying each constraint of a social organization individually, because in a real system it is unfeasible to exhaustively list all constraints (i.e. there are simply too many possibilities that have a considerable effect). Instead of modeling all possible constraints individually, we infer their combined effect from the observed social organization under a network representation. Indeed in real organizations, such constraints can be deduced from studying how individuals have to interact. Our method deduces the combined effect of these constraints by determining interaction biases among the individuals.

While I am not completely convinced by the null model, the authors addressed precisely the point I raised and I thank them for such a modification. 

This is a very good and educated observation, but we are still convinced that our measure indicates exactly what we should see for this network when considering the co-attendances only. Our method was developed for single-layer interaction networks, and with the projection we retrieve such a network. For the bipartite structure other papers indeed identified two blocks, but there are as well four large social events that were visited by women from both groups. For the single-layer interaction network we suspect that these four large social events weaken the constraint from the groups on the co-attendances of the women, and then overall the interaction network is rather homogeneous and dense. Still, the measured potentiality at 0.89 is high, but it is still not at its maximum value 1, i.e. there are constraints present in the network that restrict the interactions (although less than for the other two examined empirical data sets). We have added this comment in the text and references for the block structure present in the bipartite network.

I continue to strongly disagree: in the networks there are women that attended just two events and other that attended almost one half of them. An overlap not taking into account such activity is losing information regarding the whole system. There are various methods intended to project the information in a bipartite network into a monopartite network [1,2,3] and the naïve projection, in my view, is too sloppy to convey interesting information. 

Please, in the next versions of your paper, address the points I raised. It is in your interest.

Best regards

[1] Tumminello, M., Miccichè, S., Lillo, F., Piilo, J. & Mantegna, R. N. Statistically validated networks in bipartite complex systems. PLoS ONE 6, e17994 (2011).
[2] Gualdi, S., Cimini, G., Primicerio, K., Di Clemente, R.
& Challet, D. Statistically validated network of portfolio overlaps and systemic risk. Sci. Rep. 6, 39467 (2016).
[3] Saracco, F. et al. Inferring monopartite projections of bipartite networks: an entropy-based approach. New J. Phys. 19, 053022 (2017).